# Optimal Location of Energy Dissipation Box in Long Distance and High Drop Gravitational Water Supply System

**Weixiang Ni, Jian Zhang * and Sheng Chen ***

College of Water Conservancy and Hydropower Engineering, Hohai University, Nanjing 210098, China;
wxni917@hhu.edu.cn
* Correspondence: jzhang@hhu.edu.cn (J.Z.); chensheng@hhu.edu.cn (S.C.)

**Abstract:** In the long-distance and high-drop gravitational water supply systems, the water level difference between the upstream and downstream is large. Thus, it is necessary to ensure energy dissipation and pressure head reduction to reduce the pipeline pressure head. The energy dissipation box is a new type of energy dissipation and pressure head reduction device, which is widely used in the gravitational flow transition systems. At present, there is still a dearth of systematic knowledge about the performance of energy dissipation boxes. In this paper, a relationship between the location of the energy dissipation box and the pressure head amplitude is established, a theoretical optimal location equation of the energy dissipation box is derived, and numerical simulations using an engineering example are carried out for verification. The protective effects of an energy dissipation box placed at the theoretical optimal location and an upstream location are compared. The results indicate that for the same valve action time, the optimal position allows effectively reducing the total volume of energy dissipation box. The oscillation amplitudes of the water level in the box and the pressure head behind the box are markedly reduced. Under the condition that the water level oscillation of the energy dissipation box is almost the same, the optimal location offers better pressure head reduction protection performance than the upstream location.

**Keywords:** gravitational flow systems; energy dissipation box; optimal location; theoretical analysis; pressure head fluctuation

## 1. Introduction

The severe shortage of water resources is a critical issue worldwide, preventing accelerated urbanization and concentration of population [1,2]. Construction of long-distance water supply systems, transporting water from an area rich in water resources to another area with a shortage of water, is the most effective approach [3]. Such systems are mainly divided into those utilizing pumping station pressurization or gravity flow. Compared to the pumping station pressurization systems, gravity flow systems are advantageous where sufficient elevation difference is available to permit water to flow from a high level upstream reservoir to a low level downstream reservoir by gravity of the water supply system in the required quantity and with the required pressure head [4].

In a long-distance and high-drop gravitational water supply system (LHGWSS), the pipeline is long, the water level difference between upstream and downstream reservoir is large, and the water head in front of the valve is high. If the pipeline is not equipped with protective devices, although large amounts of energy can be consumed by the pipeline hydraulic loss by selecting a smaller pipe diameter [5], this will increase the hydrostatic pressure head in the downstream pipeline when the water delivery system is shut down. At the same time, rapid changes in flow rate can cause serious problems in water distribution systems. A high water hammer pressure head can permanently deform or rupture a pipeline and its components [6,7]. Consequently, all these factors necessitate the use of protective devices, such as pressure relief valves (PRVs), surge tanks in combination with PRVs, and valves in combination with energy dissipation boxes (EDBs).

The PRV, the important component of a hydraulic system, is usually installed at the inlet of a water supply system or the highest-pressure head point. Its working principle is that when the pressure head in front of the valve exceeds a predetermined value, the valve opens and releases some of the high-pressure water. When the pressure head in front of the valve decreases, the valve automatically closes [8–10]. Because the outlet of the PRV is connected to the tank, the pressure head drop of the PRV is frequently equal to that of the inlet pressure head [11]. However, using a PRV must be done with caution as an inappropriate selection can worsen the system transient response [12]. Unstable operation of the PRV can cause great harm, such as a decrease in discharge capacity, insufficient constant pressure head regulation, overpressure protection, resonance of the whole pipeline system, or serious damage to the sealing surface of the PRV [13,14]. There is also a rich body of research on the design and application of the water hammer protective devices using a PRV and surge tank combination. The transient flow behavior in the presence of a surge tank and PRV was investigated in the penstock of the hydroelectric power plants, where the protective devices can effectively decrease the pressure head rise within the spiral case and the turbine overspeed [15,16]. A variety of pressure head reducing valve programs were analyzed for multi-level pressure head reduction in a LHGWSS [17–20]. The results show that a pressure head reduction constant pressure head valve and surge tank could effectively eliminate the surplus water head, and ensure the safe operation of a pipeline.

The EDB investigated in the study is a new type of energy dissipation and pressure head reduction device, in which an energy dissipation valve (EDV) and an EDB are combined. Water flows through the valve structure into the energy dissipation well, which provides multi-stage energy dissipation. Compared to the PRVs and surge tanks in combination with PRV, which rely solely on the valve energy dissipation, the EDB can eliminate a large water head, reduce the working pressure head of the regulating valve in front of the box, and avoid the long-term operation of the regulating valve with a small opening. The valve operation should be stable, otherwise, when there is a problem with the PRV, the safety of the pipeline cannot be guaranteed. Depending on the local loss at the valve, the EDV in front of the box plays the role of energy dissipation and pressure head reduction, and its pressure head reduction principle is different from that of PRV. Therefore, it is suggested a combination of EDV and EDB should be adopted in the LHGWSSs. The rule of valve for the hydraulic transition process of the EDB has been fully investigated [21,22], but there is still a lack of systematic theoretical analysis of the optimal location. Most of the previous EDB locations are influenced by pipeline layout and built-in designated places; this method cannot provide an accurate basis for the reduction of water hammer in the EDB. Therefore, motivated by the above shortcomings, the optimal location for the EDB and the influence of that location on water hammer protection in an LHGWSS are investigated in this study.

The key innovative features of this paper are as follows: (1) derivation of the oscillation period and amplitude of the water level in the EDB; (2) analysis of the relationship between location of EDB and amplitude of water level; (3) derivation of theoretical equation for optimal location of EDB; (4) comparison of the pressure head oscillation under different locations.

## 2. Mathematical Model of EDB

In a LHGWSS, the working mode of combined EDV and EDB is adopted. The valve body shape is used to control the location of cavitation occurrence to ensure that the high-speed water flow will not cause damage to the valve and the EDB. The EDB is mainly composed of a folding stilling board, energy dissipation partition, energy dissipation bottom sill, and rear riprap. The water flow energy is mainly dissipated by mixing, turbulence, and breaking. Energy dissipation is achieved by exchanging mass, energy, and momentum between water and the box, gas, and itself through friction.

From the structural arrangement of the EDB, it is noted that the part of folding stilling board and energy dissipation partition consumes most of the flow energy through hydraulic friction. The second part of bottom sill and rear riprap occupies most of the volume of EDB, is mainly used to adjust the flow pattern and stabilize the flow velocity, and has the property of a surge chamber. Therefore, the EDB can be simplified as a series combination of a hydraulic friction element and an ordinary surge chamber element, and the friction element can be treated as a virtual valve with a constant opening. Its structural diagram of mathematical model is shown in Figure 1.

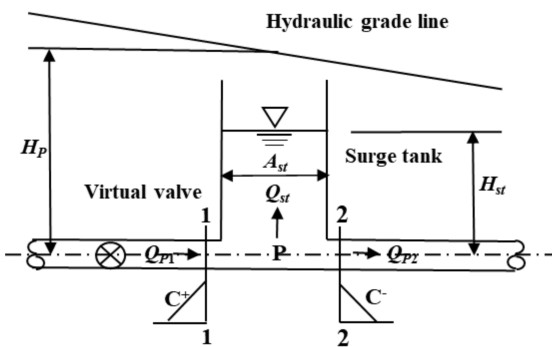

**Figure 1.** Schematic diagram of a mathematical model of energy dissipation boxes (EDB).

In Figure 1, the boundary nodes of the inlet and outlet pipes are numbered 1 and 2, respectively, and the governing equations of the hydraulic node can be expressed as follows:

Compatibility equations:

$$H_P = C_{P1} - B_{P1}Q_{P1} \tag{1}$$

$$H_P = C_{M2} + B_{M2}Q_{P2} \tag{2}$$

Relationship between the discharge and the water level:

$$\frac{dH_{st}}{dt} = \frac{Q_{st}}{A_{st}} \tag{3}$$

Head balance equation:

$$H_P = H_{st} + R_k Q_{st}|Q_{st}| \tag{4}$$

Continuity equation:

$$Q_{P1} = Q_{st} + Q_{P2} \tag{5}$$

where $H_{st}$ and $A_{st}$ are the water level and cross-sectional area of the EDB; $Q_{st}$ is the flow in/out of the impedance hole of the EDB, $Q_{st}$ is positive when water flows in; $R_k$ is the loss coefficient of impedance head, where for the components of an ordinary surge chamber, $R_k = 0$; $H_P$ and $Q_{P1}$ and $Q_{P2}$ are the transient water head and transient flow at the pipeline boundaries.

When calculating the water hammer, $\Delta t$ is small, $R_k = 0$, and Equations (3) and (4) can be simplified as follows:

$$H_{st} = H'_{st} + \frac{0.5\Delta t(Q_{st} + Q'_{st})}{A_{st}} \tag{6}$$

$$H_P = H_{st} \tag{7}$$

where $H'_{st}$ and $Q'_{st}$ are the values obtained from the previous iteration of $H_{st}$ and $Q_{st}$, respectively.

From Equations (1), (2) and (5)–(7), we can obtain:

$$H_P = \left( \frac{C_1}{C_2} + \frac{C_{P1}}{B_{P1}} + \frac{C_{M2}}{B_{M2}} \right) / \left( \frac{1}{C_2} + \frac{1}{B_{P1}} + \frac{1}{B_{M2}} \right)$$
$$C_1 = H'_{st} + 0.5\Delta t \frac{Q'_{st}}{A_{st}}$$
$$C_2 = 0.5 \frac{\Delta t}{A_{st}}$$

(8)

Variable $H_P$ can be calculated from Equation (8) and the other transient parameters can subsequently be obtained.

## 3. Theoretical Equation for Optimal Location of EDB

In a LHGWSS, when the upstream water level is high, the downstream water level is low, and the pipeline has zero flow, the pressure pipe head along the pipeline is the highest. If the hydrostatic pressure head is close to or exceeds the pipe pressure head standard, it is necessary to use an EDB to reduce the excessive water head. After the EDB is installed, the following two requirements should be satisfied: (1) there is no negative pressure head along the pipeline under the normal working conditions, and (2) the hydrostatic pressure head along the pipeline does not exceed the pipeline pressure head standard. As the pipelines of actual gravitational flow transition systems are complicated with up and down sections, the location of EDB is analyzed theoretically using a simplified pipeline.

### 3.1. Relationship Between Location of EDB and Amplitude of Water Level

The working principle of the EDB is shown in Figure 2. It is noted that the EDB is installed in the middle of the pipeline. The maximum static head in each pipeline is the maximum topographic height difference between the upper/lower reservoirs and the energy dissipation devices, and the static pressure head of the pipeline is greatly reduced. The static pressure head without energy dissipation facility is $H$, and the pressure head oscillation value is $\Delta H$. After installation of the energy dissipation device, the pipeline is divided into two sections, and the static pressure head at the end of the lower section is $H'$, m, while the pressure head oscillation value is $\Delta H'$, m. The pipeline length in front of the EDB is $L_1$, m, and the pipeline behind the EDB is $L_2$, m. $Z_1$—pipeline static pressure head when EDB is not installed, m; $Z_2$—pressure pipeline head when EDB is not installed at design flow rate, m; $Z'_1$—pipeline static pressure head when EDB is installed, m; $Z'_2$—pressure head at the end of pipeline when EDB is installed at design flow rate.

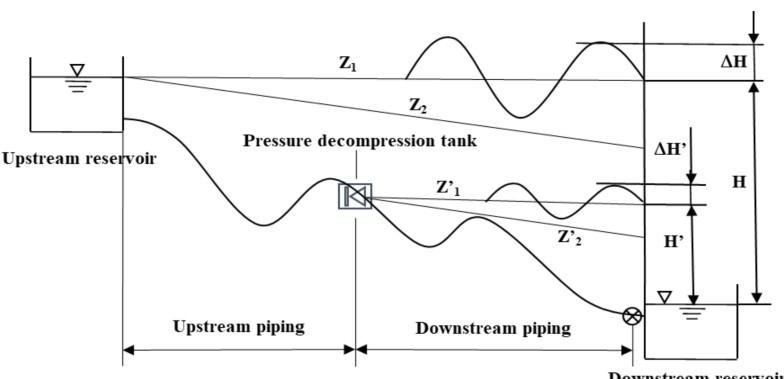

**Figure 2.** Schematic diagram of working principle of the EDB.

Assuming water incompressibility and using the harmonic vibration theory, a mathematical model of the system can be obtained as follows:

Dynamic equation:

$$H_{st} = h_w + h_j + \frac{L_2}{g} \frac{dv}{dt}$$

(9)

Continuity equation:

$$Q = vf + A_{st}\frac{dH_{st}}{dt} \tag{10}$$

where $h_w$ and $h_j$ are the head loss along the pipeline and the local head loss, respectively; $v$ is the pipeline flow velocity; $f$ is the pipe cross-sectional area; $Q$ is the pipe flow in front of the EDB; $g$ is the gravity acceleration; and $t$ is the time.

In this study, in order to directly reflect the influence of the parameters of the EDB on the water hammer pressure head caused by closing the end valve, it is assumed that the regulating valve in front of the box remains stationary and the friction of the pipeline system is ignored. Therefore, Equation (9) can be rewritten as follows:

$$H_{st} = \frac{L_2}{g}\frac{dv}{dt} \tag{11}$$

In order to simplify calculations, it is assumed that the change of flow occurs directly behind the EDB and the influence of the pipeline behind the box is ignored. The most unfavorable situation of the water level fluctuation of the EDB is that the flow in the pipe behind the box stops quickly and completely, i.e., the end valve closes quickly, $Q = 0$. Combining Equations (10) and (11), one can obtain the following:

$$\frac{d^2H_{st}}{dt^2} + \omega^2 H_{st} = 0 \tag{12}$$

where $\omega = \sqrt{gf/L_2 A_{st}}$. Equation (12) is a simple harmonic vibration equation. By introducing the initial parameters of the gravitational flow transition system, the oscillation period and amplitude of the water level in the EDB can be obtained as follows:

$$\begin{cases} T = \frac{2\pi}{\omega} = 2\pi\sqrt{\frac{L_2 A_{st}}{gf}} \\ \Delta H_{st} = v_0\sqrt{\frac{L_2 f}{g A_{st}}} \end{cases} \tag{13}$$

Equation (13) was derived under the assumption that the regulating valve in front of the box is inactive and the elasticity and friction resistance of the water in the pipeline are ignored. It is known that the shorter the distance between the EDB and the lower reservoir, $L_2$, the smaller the water hammer pressure head generated for the same valve closing time and the smaller the oscillation period. The pressure head fluctuation intensity of the downstream pipeline is reflected in Equation (13), i.e., when the EDB is close to the downstream reservoir, the pressure head fluctuation along the downstream pipeline decreases. The installation of the EDB on the pipeline can effectively reduce the static pressure head of the pipeline behind the box and the pressure head rise of the pipeline water hammer, which is conducive to the safe operation of the pipeline and reduction of maintenance cost. Therefore, for the LHGWSSs, the location of the EDB should be as close as possible to the lower reservoir.

In addition, for the LHGWSSs, the smaller the cross-sectional area of the EDB, the faster the valve moves and the severer the water level fluctuations in the box. Therefore, in order to avoid overflow or leakage from the EDB, numerical simulations should be used to calculate the reasonable size and parameters of the EDB and the valve operation rules of the gravitational flow transition system.

### 3.2. Theoretical Equation for Optimal Location of EDB

Figure 3 shows the schematic diagram of EDB location in a LHGWSS. Variables $\Delta H_A$ and $\Delta H_C$ denote the internal water pressure head at points A and C, respectively, $L_{AC}$ is the length of the pipe between points A and C, and $\alpha$ and $\beta$ are the slope angles of the hydraulic line and the angle between pipe points A and C, respectively.

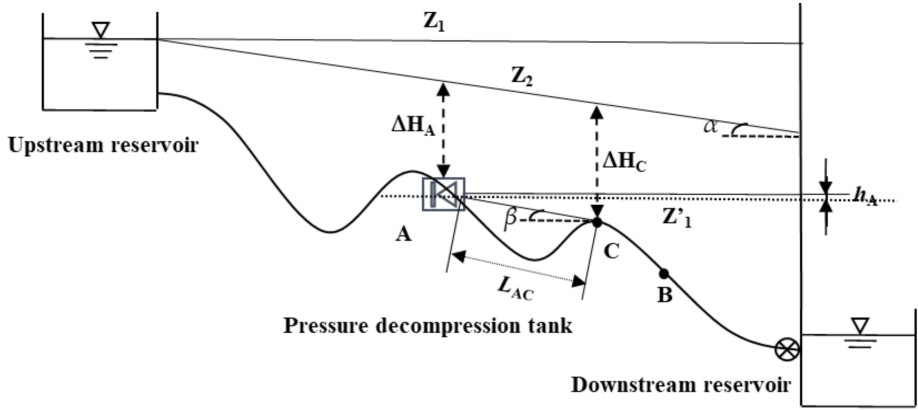

**Figure 3.** Schematic diagram of EDB location in long-distance and high-drop gravitational water supply system (LHGWSS).

As shown in Figure 3, if the pressure head along the pipeline behind point B exceeds the pipe pressure head standard, an EDB is installed at point A between the upper reservoir and point B. The relationship between point A and point C satisfies the following condition:

$$\Delta H_A = \Delta H_C - L_{AC}\frac{sin(\beta - \alpha)}{cos\alpha} \qquad (14)$$

In order to meet the overcurrent capacity of the pipeline, $\Delta H_C$ should satisfy the following condition:

$$\Delta H_C \geq \Delta H_A - h_A \qquad (15)$$

As can be seen from Equation (15), in order to avoid negative pressure head along the pipeline after installing the EDB, point C should be the highest point of the centerline elevation between the EDB and the lower reservoir (overcurrent point). For a given pipeline system, $\Delta H_C$ is a fixed value.

Combining Equations (14) and (15), one can obtain the following:

$$h_A \geq -L_{AC}\frac{sin(\beta - \alpha)}{cos\alpha} \qquad (16)$$

For a long-distance water delivery system, $L_{AC} \gg h_A$, and due to pipeline loss, $\alpha \in (0º, 90º)$, $cos\alpha \in (0,1)$, and therefore, the condition of Equation (16) can be expressed as follows:

$$sin(\beta - \alpha) \geq 0 \qquad (17)$$

Equation (17) is the theoretical equation for the location of EDB. Considering the topographic trend of the actual gravity flow system, $\beta \in (-90º, 90º)$, therefore, when $\beta$ satisfies $\beta \in [\alpha, 90º)$ and $\Delta H_A \leq \Delta H_C$, the centerline elevation at the EDB location should be higher than or equal to the centerline elevation of the overcurrent point C. On the contrary, when $\beta \in (-90º, \alpha)$ and $\Delta H_A > \Delta H_C$, the centerline elevation of the EDB location should be lower than that of the overcurrent point C. Combined with Equation (16), the lowest water head of the EDB can be obtained as follows:

$$[h_A]_{min} = \begin{cases} \Delta H_A - \Delta H_C = \frac{-L_{AC}sin(\beta-\alpha)}{cos\alpha}, & \beta \in (-90º, \alpha) \\ Const, & \beta \in [\alpha, 90º) \end{cases} \qquad (18)$$

It can be noted from Equation (18) that when $\beta \in (\alpha - 90º, \alpha)$, the lowest head of the EDB increases with the decrease in $\beta$. In order to meet the overcurrent capacity at point C, it is necessary to install an EDB at a high water level.

When $\beta \in [\alpha, 90º)$, an EDB can be installed at any water level and still ensure that there is no negative pressure head along the pipeline, and the water head of the EDB is

usually set as 3–5 m. Therefore, the EDB location should satisfy $\beta \in [\alpha, 90^\circ)$, and the overcurrent capacity of the pipeline can then also be satisfied. Additionally, the water head of the EDB is low.

It can be observed from Equation (17) that the critical location of the EDB should satisfy the following condition:

$$\beta = \alpha \tag{19}$$

When $\beta = \alpha$, angle $\beta$ of the line between points A and C is parallel to the angle of the hydraulic slope line, $\alpha$, i.e., the critical location of the EDB is the intersection of the line parallel to the hydraulic slope line and the upstream pipeline at the overcurrent point C. Under the condition that there be no negative pressure head along the pipeline, the EDB at the intersection point is closest to the downstream reservoir. Therefore, Equation (14) can be rewritten as follows:

$$\Delta H_A = \Delta H_C \tag{20}$$

Both Equations (19) and (20) are the equations for determining the optimal location of EDB. After determining the location of point C, the optimal location of an EDB can be obtained from Equation (14). After determining the water depth of the EDB, the pressure head along the pipeline behind the box decreases, there is no negative pressure head along the line, and there is a certain margin for the internal pressure head at the high point.

After determining the location and water head of the first EDB, the maximum internal pressure head of the gravity flow system should satisfy the following condition:

$$[\Delta H]_{max} < [\Delta H_S]_{max} \tag{21}$$

where $[\Delta H]_{max}$ is the maximum internal pressure head at any point along the pipeline, and $[\Delta H_S]_{max}$ is the pressure head standard at any point along the pipeline. If $[\Delta H]_{max} \geq [\Delta H_S]_{max}$, by analogy, the position and water head of the $n$th EDB can be determined so that the requirement of Equation (21) is met.

In summary, installing an EDB at the optimal location in a complex pipeline can effectively reduce the static pressure head and avoid the negative pressure head along the downstream pipeline. The above theoretical equations, Equations (19) and (20), can provide guidance and reference for the preliminary selection of the EDB location and determination of the water level in practical engineering applications. However, due to the large variation of pipeline layouts in practice, the actual position of the EDB should be adjusted appropriately with the help of the theoretical equation.

## 4. Influence of EDB Location on Water Transition System

A LHGWSS analyzed in this section is shown in Figure 4. The upstream and downstream water level drops of the gravity flow system pipeline are large and the slope is steep along the pipeline. The total length of the pipeline is 48.199 km, and the design flow rate is 0.35 m$^3$/s. The water level of upstream reservoir is 2017.50 m and that of the downstream reservoir is 1567.00 m. The head drop of the upstream and downstream reservoir is 450.5 m and the maximum centerline elevation difference is 449.80 m. Two flow regulating valves (DN1200 and DN800) are installed in parallel at the end of the pipe. Using the characteristic line method, a one-dimensional hydraulic transient numerical model of the water transition system is established. In this case, when closing the valves along the pipeline, there should be no negative pressure head along the pipeline, and the maximum pressure head should be lower than the pipeline pressure head standard to ensure the safe and stable operation of the system. The pressure pipe head line and pipe pressure head standard line along the pipeline under the condition of constant flow are shown in Figure 4.

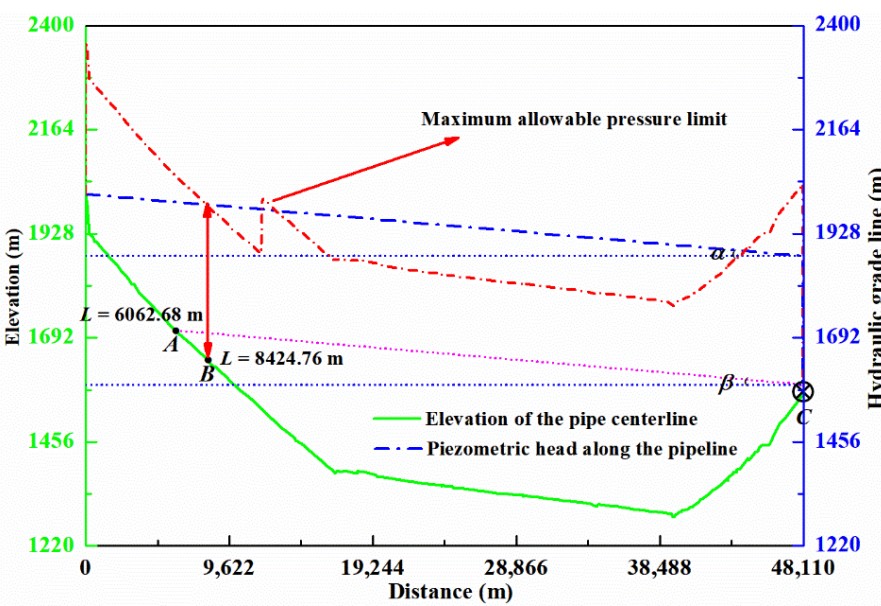

**Figure 4.** A LHGWSS of a practical project.

### 4.1. Steady Flow Analysis of Gravity Flow System without EDB

As shown in Figure 4, the hydrostatic pressure head begins to exceed the pipe pressure head standard at $L = 8424.76$ m (point B). From stake 8 + 424.76 to 11 + 753.00 and 13 + 022.28 to 43 + 957.34, the water pressure head along the pipeline is higher than the maximum allowable positive pressure head along the pipeline (stake 0 + 00.00–11 + 753.00, 350 m; and 11 + 753.00–48 + 101.18, 480 m). In particular, at stake 39 + 352.63, the pressure head difference between the hydrostatic pressure head and the maximum working pressure head reaches the maximum value (617.71 m − 480 m = 137.71 m). Therefore, it is necessary to install EDBs to reduce the pipeline pressure head of the LHGWSS and avoid extreme situations, such as pipe bursting, to ensure the safe operation of the pipeline.

### 4.2. Theoretical Analysis of Optimal Location of EDB

According to the angle of the hydraulic slope line in Figure 4, it can be found that $\sin \alpha = 0.002912$ and $\cos \alpha \approx 1$. As shown in Equation (19), the optimal location of the EDB meets the condition $\beta = \alpha$. Therefore, it can be found from the relationship between $\alpha$ and $\beta$ that $\sin \beta = 0.002912$ and $\cos \beta \approx 1$.

According to the trend of the pipeline centerline elevation analyzed in Figure 4, in order to better prevent the negative pressure head occurrence in the pipeline generated by the EDB, the overcurrent point C should be located at the high point at the end of the pipeline (stake 480.079.04 with the internal water pressure head $\Delta H_C = 167.79$ m). According to Equation (19), a theoretical straight line parallel to the hydraulic slope line is drawn between point C and the upstream pipeline. The intersection of the line and the pipeline centerline elevation (point A, stake 60.062.68) is the theoretical optimal location of the EDB, and the internal pressure head at point A is 167.76 m.

### 4.3. Comparison and Analysis of Protective Measures for Different EDBs

In this study, in order to meet the overcurrent capacity of the highest point of the downstream pipeline, the water depth of the EDB is set as 3.26 m. A DN1000 EDV is connected in series in front of the box. It can be seen from Figure 4 that the EDB is installed at point A (scheme *a*), and the hydrostatic pressure head along the pipeline is much lower than the maximum working pressure head. Additionally, there is no negative pressure head along the pipeline, which meets the overcurrent capacity requirement of the pipeline. Firstly, the protection effect of schemes a and b (the EDB is installed at stake 20.064.00) is compared for the same size and valve closing time. Secondly, the protection effect of

scheme *a* is compared to that of scheme *c* (the EDB is installed at stake 20.064.00) for different size parameters, and valve action and closing time. For the three schemes, the pressure reduction heads of the EDV are all 72.88 m. The valve opening and closing rules for the three schemes are shown in Tables 1 and 2, and the extreme pressure head along the pipeline for the different schemes is shown in Table 3. The parameters of the EDB and the bottom pressure head for the different schemes are compared in Table 4, and the protective effects of water hammer for the three schemes are shown in Figures 5–7.

As the centerline elevations at the EDB location in the two schemes differ, the EDV and box in schemes *a* and *b* jointly reduce the heads by 288.77 m and 148.20 m, respectively. The EDV in front of the box is kept fully open and the pressure head is reduced by 72.88 m, and the pressure heads of the EDBs are reduced by 215.89 m and 75.32 m, respectively. As shown in Figure 5, the initial internal pressure head in scheme *b* is higher than that in scheme *a* along the pipeline between 6062.68 m and 9915.76 m, while between 9915.76 m and the end of the pipeline, the initial internal pressure head in scheme *a* is lower than that under scheme *b*, because the EDB in scheme *a* reduces the water heads more.

**Table 1.** Valve opening and closing rules for schemes a and b.

|          | Valve Diameter | Opening Degree | Action Time | Valve Closing Slope | Valve Closing Time |
|----------|----------------|----------------|-------------|---------------------|--------------------|
|          | DN1000         | 1              | 120 s       | 1/400               | 400 s              |
| scheme a | DN1200         | 0.9767         |             | 1/600               | 585 s              |
|          | DN800          | 0.6928         | 0 s         |                     | 415 s              |
|          | DN1000         | 1              | 120 s       | 1/400               | 400 s              |
| scheme b | DN1200         | 0.5520         |             | 1/1062              | 585 s              |
|          | DN800          | 0.4037         | 0 s         | 1/1030              | 415 s              |

**Table 2.** Valve opening and closing rules for scheme *c*.

| Valve Diameter | Opening Degree | Action Time | Break Point | Two Broken Line Slope | Actual Closing Time | Total Time |
|----------------|----------------|-------------|-------------|-----------------------|---------------------|------------|
| DN1000         | 1              | 120 s       | 0           | 1/800                 | 800 s               | 800 s      |
| DN1200         | 0.5520         |             |             | 1/1782                | 360 s               | 1500 s     |
|                |                |             |             | 1/3257                | 1140 s              |            |
|                |                | 0 s         | 0.35        | 1/6705                | 360 s               | 1500 s     |
| DN800          | 0.4037         |             |             | 1/3257                | 1140 s              |            |

**Table 3.** Extreme pressure head along pipeline for different schemes.

| Scheme   | Maximum Pressure Head before Box | Minimum Pressure Head before Box | Maximum Pressure Head behind Box | Minimum Pressure Head behind Box |
|----------|----------------------------------|----------------------------------|----------------------------------|----------------------------------|
| scheme *a* | 315.92 m                       | 2.3 m                            | 443.15 m                         | 2.12 m                           |
| scheme *b* | 155.53 m                       | 1.31 m                           | 590.35 m                         | 3.26 m                           |
| scheme *c* | 154.95 m                       | 1.35 m                           | 581.14 m                         | 3.16 m                           |

**Table 4.** Comparison of EDB parameters and bottom pressure head for different schemes.

| Scheme   | Initial Water Depth | Cross-Sectional Area | Highest Water Depth | Lowest Water Depth | Bottom Maximum Pressure Head | Bottom Minimum Pressure Head |
|----------|---------------------|----------------------|---------------------|--------------------|------------------------------|------------------------------|
| scheme *a* |                     |                      | 3.72 m              | 1.99 m             | 3.72 m                       | 1.99 m                       |
| scheme *b* | 3.26 m              | 25 m$^2$             | 7.37 m              | 3.26 m             | 7.37 m                       | 3.26 m                       |
| scheme *c* |                     | 150 m$^2$            | 3.78 m              | 2.13 m             | 3.78 m                       | 2.13 m                       |

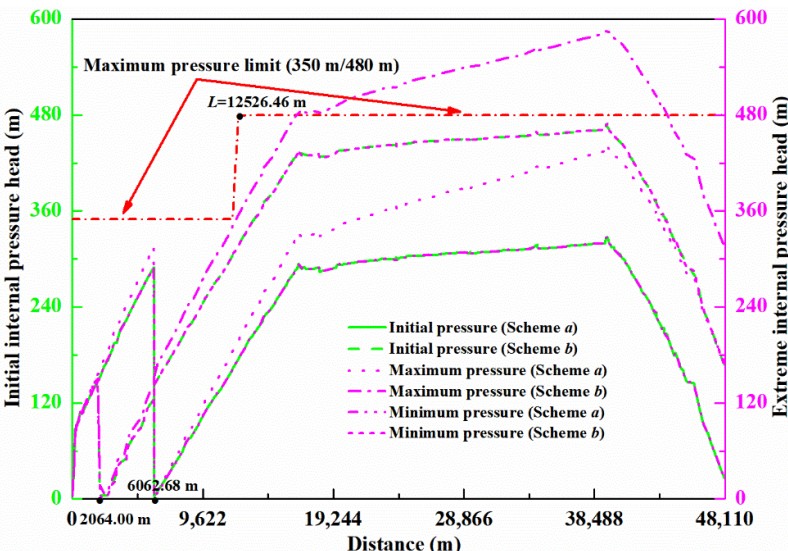

**Figure 5.** Influence of different locations of EDB on maximum and minimum pressure head in pipelines (schemes *a* and *b*).

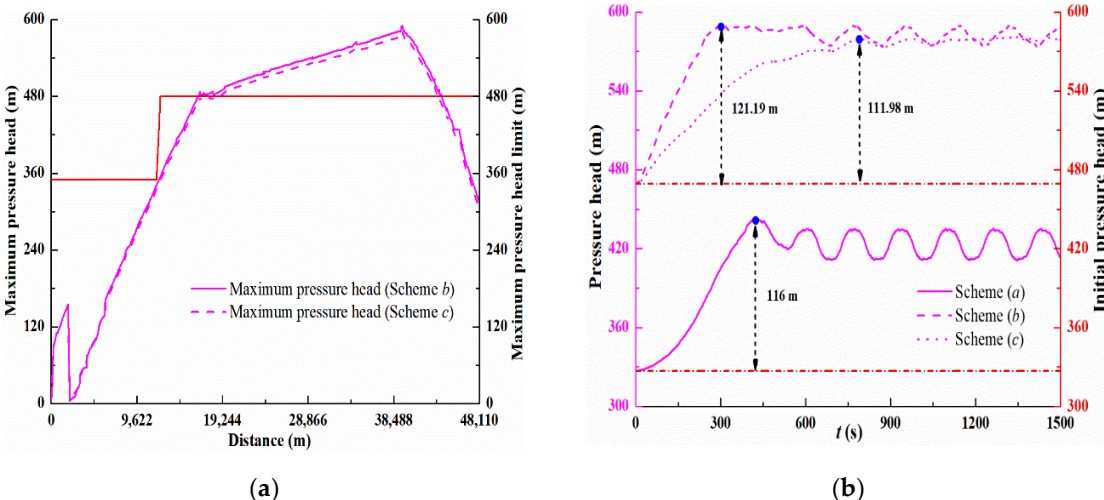

(**a**)                    (**b**)

**Figure 6.** (**a**) Pressure head change along the pipeline in schemes b and c; (**b**) pressure head change at maximum pressure head point behind energy dissipation box in schemes a, b, and c.

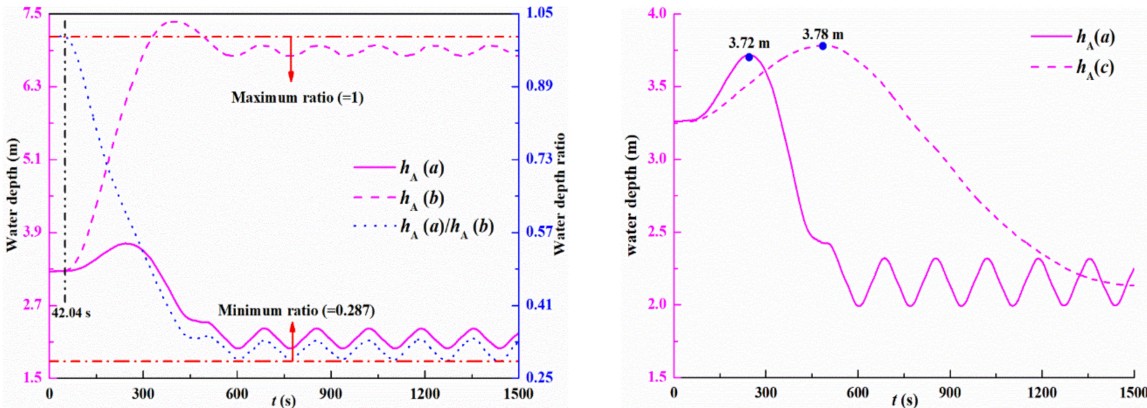

**Figure 7.** (**a**) Influence of EDB location on water depth and water depth ratio of the energy dissipating box (schemes *a* and *b*); (**b**) water depth change of the energy dissipating box in schemes a and b.

As listed in Table 3, for the same size of EDB, the valve action and closing time of the maximum pressure heads of schemes *a* and *b* before the box are 315.92 m and 155.53 m, respectively, while the maximum pressure heads behind the box are 443.15 m and 590.35 m, respectively. In scheme *a*, the difference between the maximum internal pressure head along the pipeline and the pipeline pressure head standard is ∣443.15 m − 480 m∣ = 36.85 m, and the maximum internal pressure head along the pipeline does not exceed the working pressure head of the pipeline. In scheme *b*, the maximum pressure head along the pipeline from 16,553 m to 43,839.25 m exceeds the pipeline pressure head standard, and the maximum pressure head difference is 110.35 m (located at $L$ = 39,410.04 m, where the pipeline centerline elevation is the lowest). This is because the EDB in scheme *b* is located near the upper reservoir, and the pressure head reduction under constant flow is 288.77 − 148.20 = 140.57 m less than that in scheme *a*. As a result, the pressure head along the pipeline for the same valve closing time is higher than the internal water pressure head in scheme *a*. In addition, there is no negative pressure head in the two schemes, which meets the overcurrent capacity requirement of the pipeline. Compared to scheme *b*, the maximum/minimum pressure head in scheme *a* is within the range of pressure head standard. Therefore, scheme *a* can provide better water hammer protection, which illustrates further the usefulness of the optimal location theory of EDB.

As shown in Table 3, the maximum pressure heads along the front pipe of the box in schemes b and c are 155.53 m and 154.95 m, respectively, and the maximum pressure heads along the pipe behind the EDB in schemes b and c are 590.35 m and 581.14 m, respectively. When the closing time of the valve increases (the closing time of the EDV is from 400 s to 800 s, for DN1200 end valve from 585 s to 1500 s, and for DN800 end valve from 415 s to 1500 s), the maximum pressure head along the pipeline in scheme c is higher than that in scheme a, but the pressure head is not significantly different.

As shown in Figure 6a, the maximum pressure along the pipeline in scheme c also exceeds the maximum working pressure of the pipe between 1912.89 m and 43,451.81 m. In Figure 6b, the maximum amplitudes of the maximum pressure head point along the pipeline behind the EDB in schemes a, b, and c are 116 m, 121.19 m, and 111.98 m, respectively. It is noted that after shutting the valve with a two-fold line, the maximum pressure amplitude of scheme c is 9.21 m lower than that of scheme b, and the pipeline pressure is alleviated, but the difference in the maximum pressure amplitude of the three schemes is very small. Therefore, in order to reduce further the pressure along the pipeline in scheme b, a longer valve closing time is needed. However, the longer the valve closing time, the lower the water level in the box, which entails higher requirements for the size of the EDB and increases the project cost. The initial internal pressure head along the back of the EDB in schemes b and c is higher than that in scheme a, and the maximum static pressure head difference along the pipeline is 142 m. Therefore, even if the valve closing time is further extended, the maximum pressure head along the pipeline will still exceed the pipeline pressure standard during valve closing in schemes b and c.

Figure 7 shows the change in water depth in the EDB for different locations. Due to the influence of valve action time, the water level of the EDB rises first and then decreases in the two schemes. As listed in Table 4, the maximum water depth of the EDB in schemes a and b is 3.72 m and 7.37 m, respectively, and the lowest water depth is 1.99 m and 3.26 m, respectively. In the half length of scheme *a* ($t = L_2/c$ = 42,038.5/1000≈42.04 s), the water depth ratio of the EDB remains unchanged (=1) in the two schemes. From $t$ = 42.04 s, the downstream positive wave travels to the bottom of the EDB, and the water depth of the EDB in scheme a begins to fluctuate, resulting in a change in the water-depth ratio. The EDB location in scheme a is closer to the lower reservoir than in scheme b, i.e., distance $L_2$ is shorter. From Equation (13), it is noted that the closer the EDB is to the downstream reservoir, the smaller the water level fluctuation in the box and the larger the oscillation period. The amplitude of the water level in the EDB in scheme a is smaller than that in scheme b, as shown in Figure 7a, and the fluctuation period of the water level in scheme *a* is longer before the valve is completely closed. The theoretical analysis is consistent with the

results of numerical simulations. After 42.04 s, the water depth ratio in the box continues to decrease until it stabilizes, and the minimum water depth ratio is 0.287. From the above analysis, it can be concluded that for the same valve closing rule, the water level fluctuation of the EDB in scheme *b* is severer than in scheme *a* and the requirements for the size of the EDB are higher.

As shown in Table 4, the maximum water depth of the EDB in schemes *a* and *c* is 3.72 m and 3.78 m, respectively, the lowest water depth is 1.99 m and 2.13 m, respectively, and the cross-sectional area of the EDB in schemes *a* and *c* is 25 m$^2$ and 150 m$^2$, respectively. Therefore, after increasing the cross-sectional area of the EDB and increasing the valve closing time, the amplitude of the water level of the EDB located at 20.064.00 is not much different from that located at 60.062.68, as shown in Figure 7b. It is noted from Equation (13) that the water level oscillation period is proportional to the cross-sectional area of the EDB, and the larger the area, the longer the oscillation period. The cross-sectional area of the EDB is inversely proportional to the water level amplitude: the larger the area, the slower the speed of the water body in the pressurized pipeline flowing in and out of the EDB, and the more stable the water level amplitude in the box. Compared to scheme *b*, the maximum water depth in scheme *c* decreased by 3.59 m. Therefore, in order to meet the requirements of alleviating the water level fluctuations of the EDB located at the front of the pipeline, it is necessary to increase the cross-sectional area of the EDB. However, at the same time, under the same requirements of water level amplitude protection, locating the EDB before the optimal location increases the project cost.

## 5. Conclusions

The following conclusions can be drawn in this paper:

1.  A simplified oscillation period and amplitude equation of the water level in the EDB, under the assumption of the EDV, remains stationary and the friction of the pipeline system could be negligible.
2.  With the premise that no negative pressure head along the pipeline under the normal working conditions and the hydrostatic pressure head along the pipeline does not exceed the pipeline pressure head standard, the equations for determining the optimal location of EDB are proposed and derived.
3.  The theoretical optimal location performs better than the upstream location for water hammer protection. The initial internal pressure head of optimal location is lower than that of upstream location inasmuch as the effect of reducing the water head. Under the almost same water level oscillation against safety margin, the theoretical optimal location can effectively reduce the total volume and the valve closing time compared with the upstream location, thereby saving in project investment. Additionally, given a fixed air vessel volume and valve closing time, the optimal location provided better protection performance against the positive pressure head of the water hammer compared to the upstream location. Furthermore, the maximum water depth is much lower with the optimal location.
4.  This study provides theoretical and numerical support for selecting the optimal location for an EDB in LHGWSS. However, it must be noted that, in this study, the practical project introduced in this paper is still under construction, so no measured data are available to make comparison with the results. Therefore, future research will be carried out to further verify the rationality of the optimal location for an EDB after the measured data are next obtained in the future.

**Author Contributions:** Conceptualization, W.N. and J.Z.; methodology, W.N.; software, W.N.; validation, J.Z. and S.C.; formal analysis, W.N. and J.Z.; investigation, W.N.; writing—original draft, W.N.; writing—review and editing, J.Z. and S.C. All authors have read and agreed to the published version of the manuscript.

**Funding:** This research was funded by the Fundamental Research Funds for the Central Universities (grant number: 2017B690X14), and the Postgraduate Research & Practice Innovation Program of

Jiangsu Province (grant number: KYCX17_0435), the National Natural Science Foundation of China (grant number: 51709087).

**Institutional Review Board Statement:** Not applicable.

**Informed Consent Statement:** Not applicable.

**Data Availability Statement:** Data sharing not applicable.

**Conflicts of Interest:** The authors declare no conflict of interest.

## Abbreviations

| | |
|---|---|
| LHGWSS | Long-distance and high-drop gravitational flow transition system |
| PRV | pressure relief valve |
| EDB | energy dissipation box |
| EDV | energy dissipation valve |

## Abbreviations

The following symbols are used in this paper:

| | |
|---|---|
| $A_{st}$ | cross-sectional area of EDB ($m^2$); |
| $f$ | pipe cross-sectional area ($m^2$); |
| $g$ | gravity acceleration, 9.8 $m/s^2$; |
| $H_P$ | pressure head at boundary point P of EDB (m); |
| $H_{st}$ | water level of EDB (m); |
| $H_{st}'$ | known water level at time of $t—\Delta t$ (m); |
| $h_A$ | water depth, (m); |
| $h_w$ | head loss along pipeline (m); |
| $h_j$ | local head loss (m); |
| $L_{AC}$ | length of pipe between points A and C (m); |
| $Q$ | pipe flow in front of EDB ($m^3/s$); |
| $Q_{P1}$ | discharge into junction P ($m^3/s$); |
| $Q_{P2}$ | discharge out of junction P ($m^3/s$); |
| $Q_{st}$ | flow in/out of impedance hole of EDB ($m^3/s$); |
| $Q_{st}'$ | known discharge at time of $t—\Delta t$ ($m^3/s$); |
| $R_k$ | loss coefficient of impedance head ($s^2/m^5$); |
| $t$ | time (s); |
| $v$ | pipeline flow velocity (m/s); |
| $\alpha$ | slope angle of hydraulic line (º); |
| $\beta$ | angle between pipe points A and C (º); |
| $\Delta t$ | time difference (s); |
| $\Delta H_A$ | internal water pressure head at point A (m); |
| $\Delta H_C$ | internal water pressure head at point C (m); |
| $[\Delta H]_{max}$ | maximum internal pressure head at any point along pipeline (m); |
| $[\Delta H_s]_{max}$ | pressure head standard at any point along pipeline (m). |

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
