# Peer review of "Optimal Location of Energy Dissipation Box in Long Distance and High Drop Gravitational Water Supply System"

_water, doi:10.3390/w13040461_

Round 1

Reviewer 1 Report

Title: It is not clear what is the exact meaning of “energy dissipation box” and “high drop gravitational”

Abstract: Do you think rather pressure release or reduction under the term “energy dissipation”? When you use “decompression device” do you refer to release of gas from water? Which gas? If it is not about gas, it is not decompression. Upstream and downstream are something else in oil and gas engineering, so please clarify. Do you mean pumping of water from lower level to higher or opposite in meaning gravitational flow?

Introduction: Please be more focused and try to make this section shorter.

Main text: Figure 1 and 2 are similar. Please use only one of them. If you use term pressure, you should use pressure units such bar and Pa, while for head (pressure head) you can use unit of distance, i.e. meter. Please supply your text with nomenclature or notation (before references), where you will list all used variables, constants with their units (also meaning of indexes). 

Figures: All figures must be better prepared.

References: Where available, please add doi links.

Reviewer 2 Report

This paper is about the energy dissipation box which is a new type of energy dissipation and decompression device, which is widely used in the gravitational flow transition systems. A relationship between the location of the energy dissipation box and the pressure amplitude is established, a theoretical optimal location equation of the energy dissipation box is derived, and numerical simulations using an engineering example are carried out for verification. The protective effects of an energy dissipation box placed at the theoretical optimal location and an upstream location are compared
The research design is appropriate. The methods and results are adequately described. The results are clearly presented. The conclusions are supported by the results.

The notations and formulas in the text are not the right sizes in the paper.
Figure 1, 2. Make the figures well arranged.
Tables 1-4. are not positioned correctly in the pdf.
Page 15. Position all the descriptions of the abbreviations correctly!

Reviewer 3 Report

The authors are commended for contributing to the complex analysis and behaviors of pressure relief and energy dissipation elements in gravitational water supply systems.

The manuscript is generally well-worded and has a logical presentation that makes it easy to read. The authors reviewed past and recent studies and focused their contribution mainly to deriving theoretical relationships for the location of an energy dissipation box.

I find the publication to be premature and somehow shallow in content. The overall content is not seminal as it does not provide any new information on the topic to the scientific and the engineering community at this stage.

It is recommended that the authors consider including measured data/information from their on-going experiments (which they have mentioned is under construction) to validate the theoretical results. A detailed validation/verification is required to make the results meaningful. The authors have indicated that they performed ‘verification’. In my view, the term ‘verification’ is improperly used in the paper as written and presented. One would have used the word ’evaluation’ instead. In an engineering definition, ‘validation/verification’ is a process of comparing the model and its behavior to the real system and its behavior and not with simulated data (which may or may not represent the real system).

Round 2

Reviewer 1 Report

No further comments.

Reviewer 3 Report

The authors did not provide any response to the recommendation that   measured data/information from their on-going experiments be included to validate the theoretical results.